# Effect of Cloth Masks and N95 Respirators on Maximal Exercise Performance in Collegiate Athletes

**DOI:** 10.3390/ijerph19137586

**Published:** 2022-06-21

**Authors:** Matthew E. Darnell, Tyler D. Quinn, Sean P. Carnahan, Tyler Carpenter, Nicholas Meglino, Patrick L. Yorio, Jeanne M. Doperak

**Affiliations:** 1Department of Sports Medicine and Nutrition, School of Health and Rehabilitation Sciences, University of Pittsburgh, 3860 South Water Street, Pittsburgh, PA 15203, USA; nim114@pitt.edu; 2National Personal Protective Technology Laboratory (NPPTL), National Institute for Occupational Safety and Health (NIOSH), Centers for Disease Control and Prevention (CDC), 626 Cochrans Mill Rd, Pittsburgh, PA 15236, USA; yhh7@cdc.gov; 3Department of Orthopedic Surgery, University of Pittsburgh Medical Center, University of Pittsburgh, 3200 South Water Street, Pittsburgh, PA 15203, USA; carnahansp@upmc.edu (S.P.C.); doperakjm@upmc.edu (J.M.D.); 4Department of Athletics, University of Pittsburgh, 3502 Aliquippa Street, Pittsburgh, PA 15261, USA; carpenter.435@osu.edu; 5Strategic Programs Office (SPO), Human Resources Office (HRO), Office of the Chief Operating Officer (OCOO), 4770 Buford Hwy, Atlanta, GA 30341, USA; wjc8@cdc.gov

**Keywords:** COVID-19, face covering, graded exercise test, personal protective equipment

## Abstract

This study compared exercise performance and comfort while wearing an N95 filtering facepiece respirator (N95), cloth mask, or no intervention control for source control during a maximal graded treadmill exercise test (GXT). Twelve Division 1 athletes (50% female, age = 20.1 ± 1.2, BMI = 23.5 ± 1.6) completed GXTs under three randomized conditions (N95, cloth mask, control). GXT duration, heart rate (HR), respiration rate (RR), transcutaneous oxygen saturation (SpO_2_), transcutaneous carbon dioxide (TcPCO_2_), rating of perceived exertion (RPE), and perceived comfort were measured. Participants ran significantly longer in control (26.06 min) versus N95 (24.20 min, *p* = 0.03) or cloth masks (24.06 min, *p* = 0.04). No differences occurred in the slope of HR or SpO_2_ across conditions (*p* > 0.05). TcPCO_2_ decreased faster in control (B = −0.89) versus N95 (B = 0.14, *p* = 0.02) or cloth masks (B = −0.26, *p* = 0.03). RR increased faster in control (B = 8.32) versus cloth masks (B = 6.20, *p* = 0.04). RPE increased faster in the N95 (B = 1.91) and cloth masks (B = 1.79) versus control (B = 1.59, *p* < 0.001 and *p* = 0.05, respectively). Facial irritation/itching/pinching was higher in the N95 versus cloth masks, but sweat/moisture buildup was lower (*p* < 0.05 for all). Wearing cloth masks or N95s for source control may impact exercise performance, especially at higher intensities. Significant physiological differences were observed between cloth masks and N95s compared to control, while no physiological differences were found between cloth masks and N95s; however, comfort my differ.

## 1. Introduction

Aligned with April 2020 CDC recommendations to facilitate source control, universal masking was adopted by many athletic associations including the National Collegiate Athletic Association (NCAA) [1] and the Pennsylvania Interscholastic Athletic Association (PIAA) [2]. The recommendations for athlete source control have often included masking during strenuous activity, times of game preparation, recovery, and in some cases, athletic competitions [1,2]. While mask mandates and guidelines for the general population and athletes have fluctuated based on community disease prevalence, hospitalization and death rates, and vaccine status over the course of the pandemic, many athletes are still wearing masks of various types during practice, training, and competitions. For example, the Medical Advisory Group for the Atlantic Coast Conference (ACC) did not mandate mask wearing during activity, but some ACC teams such as women’s volleyball during the 2020/21 season elected to wear masks for source control during competition. Similarly, the Big 10 volleyball teams also elected to wear masks during competition.

While masking in the face of disease is not a novel approach to mitigate viral spread, such widespread use, including masking during strenuous athletic activity, has an unknown effect on physiology and performance. Further complicating the problem, individuals wore a wide variety of products to achieve the desired source control included masks (often referred to as barrier face coverings, cloth masks, face masks, facial coverings, etc., in different contexts or to refer to masks made of different materials) and respirators (e.g., N95 filtering facepiece respirators (N95s)) worn for the purpose of source control. However, the differential impact of using these various types of products for source control on athletic performance is unknown. Traditionally, the research in this area has focused on the impact of wearing fit-tested N95s or other National Institute for Occupational Safety and Health (NIOSH)-approved respiratory protection devices during moderate intensity occupational tasks [3,4,5,6]. While some small physiological performance changes have been noted in these studies, none have found differences that are clinically relevant to user safety (the main concern in this type of occupational research), few were focused on performance decrements at high intensities important to athletes, and fewer still have focused on fit-tested respirators rather than on masks or respirators used as source control such as those being recommended for public use during the COVID-19 pandemic [6,7,8,9,10].

In athletics, not only is disease transmission risk and source control a concern when making mask use recommendations, exercise performance and safety implications must also be considered. While it is known from previous research that exercise performance time is linearly and negatively related to inhalation resistance (induced by respirators and masks) [3,11,12], it is still unclear how these assumptions will affect athlete performance in real-world settings or across different mask types. Since the start of the COVID-19 pandemic, only one study has looked at wearing a cloth mask during activity, concluding that cloth masks reduced graded exercise test (GXT) time by 14% compared to no mask. However, the study sample of 31 participants were not elite athletes and the study did not include a comparison to any other type of source control product, such as an N95 respirator [9]. These limitations in external validity are important to address prior to translating the results to recommendations specifically for athletes while also considering performance across mask types. Accordingly, the purpose of this study is to compare the physiological and performance effects of three source control interventions (N95, cloth mask, and no intervention) during high intensity graded exercise in elite athletes. The results will have practical implications for understanding the ability for individuals to perform high-intensity activities while wearing an N95 or cloth mask for source control.

## 2. Materials & Methods

### 2.1. Participants

Recruitment of all volunteer participants occurred through a search of scholarship division 1 athletes at the University of Pittsburgh. Initial contact was made with all potential participants via word of mouth. Athletes interested in participation attended a session where the details of the study and their participation were explained and eligibility criteria were examined. If the athletes remained eligible and interested, they were asked to sign an informed consent document prior to study initiation. Participation included a screening followed by three separate visits to the University of Pittsburgh Fitzgerald Field House Gymnasium where testing was completed across six weeks in May and June 2020. Eligible participants were (1) between 18 and 24 years of age, (2) a current NCAA athlete, and (3) free from contraindications for maximal treadmill exercise testing such as musculoskeletal injury, cardiopulmonary disease, taking medications that affect the heart rate response to exercise, or considered to be a high-risk population according to CDC COVID-19 guidelines. No financial reimbursement was provided to participants.

Sample characteristics are presented in Table 1. Participants were 50% female, had an average age of 20.1 ± 1.2 years, weight of 68.7 ± 9.1 kg, height of 1.7 ± 0.1 m, and BMI of 23.5 ± 1.6 kg/m^2^. Participants were student athletes distributed across gymnastics (*n* = 2, 16.7%), wrestling (*n* = 5, 41.7%), soccer (*n* = 4, 33.3%), and swimming (*n* = 1, 8.3%).

### 2.2. Ethical Considerations

Ethical approval was obtained from the University of Pittsburgh Human Protection Research Office (Study ID# 20080095). All participants signed an informed consent prior to study participation.

### 2.3. Graded Treadmill Exercise Test

All participants performed three separate treadmill GXTs (one for each mask condition) at the University of Pittsburgh Fitzgerald Field House Gymnasium using a motorized treadmill (Woodway, 4Front, Waukesha, WI, USA). Prior to each GXT, the participant sat quietly for 5 min (rest stage), followed by a 5-min walking period (walking stage, 5.31 km per hour (km/h) and 0% grade). After completion of the walking stage, the participant moved onto a modified Astrand Running Protocol for maximal treadmill exercise [13]. This protocol consisted of a jogging warm-up for 5 min followed by 3-min stages of constant treadmill speed and progressively higher grade (2.5% increase each stage). The jogging warm-up (80% of testing speed) and testing running speeds (10.5–16 km/h) were determined based on the participant’s self-reported 2- or 3-mile running time. Verbal motivation was used to provide positive encouragement to the participants to promote continued effort. While the motivation provided was not recorded, it was provided in an anecdotally similar way across all experimental sessions. After indicating that they had reach terminal fatigue, the participants were instructed to remove themselves from the moving treadmill to the side rails which marked the completion of the GXT. All physiological and perceptual parameters were recorded within 10 s of GXT termination to serve as the end-exercise values. After test termination, all participants performed a 5-min walking cool-down on the treadmill (5.31 km/h, 0% grade) followed by a five-minute seated recovery period.

Heart rate (HR) and respiratory rate (RR) were measured continuously throughout the exercise test using the validated Hexoskin smart shirt (Hexoskin Pro Shirt, Carre Technologies Inc., Montreal, QC, Canada) [14,15,16]. Transcutaneous oxygen saturation (SpO_2_) and transcutaneous carbon dioxide (TcPCO_2_) was measured continuously throughout the testing using a transcutaneous ear lobe sensor validated for use during graded treadmill exercise (TOSCA 500, Radiometer Medical, Denmark, Sweden) [17]. Ratings of perceived exertion (RPE) were measured on a scale from 0 (extremely easy) to 10 (extremely hard) using the validated OMNI rating of perceived exertion pictorial scale [18]. All variables were recorded manually during the last 30 s of each timed protocol stage (rest, walk, exercise, etc.). All sensors and the smart shirt were fitted to the participants using the associated manufacturer instructions for sizing, preparation, and attachment prior to beginning the measurement and exercise protocol.

### 2.4. Experimental Conditions

Each participant performed three trials which included the following source control interventions: no intervention, cloth mask, and a NIOSH-approved N95 respirator. The cloth mask was the same brand and style between participants: triple-layer 100% cotton face mask with elastic ear loops (model: #661692, Old Navy, San Francisco, CA, USA). The N95 used was the same model for all tests (1860, 3M, St Paul, MN, USA). A user seal check of the N95 was performed by each participant prior to beginning each test [19]. No formal fit tests were completed because the intent was to examine the use of the N95 as source control. After each trial, the N95s were thrown away and the cloth masks were machine washed prior to reuse. During each trial, participants were not permitted to touch or readjust their respective mask or N95. Athletes were randomly assigned and counterbalanced to one of six experimental sequences to minimize any order effect. There was a minimum 48-h wash-out period between each testing day for all participants.

### 2.5. Subjective Comfort and Wearing Experience

Following each cloth mask and N95 trial, the participants were asked to provide feedback regarding their subjective comfort as well as the perceived wearing experience using the validated Respirator Comfort, Wearing Experience, and Function Instrument (R-COMFI) [20]. While this instrument was designed for use in healthcare workers, most of the questions translate well across populations except for the questions related to function during healthcare specific work (#17–21) [20]. As such, those questions were omitted from this study.

### 2.6. Experimental Controls

Prior to each GXT, participants were asked to record nutritional intake as well as physical activity for that day. Each participant was encouraged to reproduce their intake and physical activity pattern as closely as possible prior to each trial. Further, each participant was encouraged to wear the same footwear for each of their respective trials.

### 2.7. Statistical Analysis

Descriptive statistics were used to describe the demographic characteristics of the sample population presented in Table 1. The use of linear mixed models with post-hoc Bonferroni adjustment was employed to compare the differences in GXT termination time across the three source control interventions of interest (no intervention, N95, and cloth mask) (Figure 1). Linear regression models were used to estimate the simple slope of each of the five physiological parameters measured (heart rate, respiration rate SpO_2_, TcPCO_2_, and RPE) over time during the exercise test. Further, moderation by mask type was explored by including the interaction between time and a dummy variable indicating the intervention condition (i.e., 1, 2, 3) into the regression model to test for the pairwise differences between the estimated regression slopes across the three intervention conditions. Along with the simple linear regression results, Table 2 includes the results of the moderation analysis. Table 2 shows the estimated regression coefficients for each intervention condition along with a *p* value of the difference between the prediction of termination time by intervention condition. The column labeled “*p* value of difference” indicates whether the physiological variable over time is significantly different in each intervention condition. If *p* values in this column are below 0.05, significant moderation is indicated. The variable of time was centered for the models. All models controlled for the effects of participant age, height, weight, and BMI.

Subjective responses to the R-COMFI questionnaire were summarized using descriptive statistics for the N95 and cloth mask (Table 3). Scores for each question were compared using paired sample *t*-tests. The alpha level was set at 0.05 for all comparisons and all analyses were performed in the Statistical Package for the Social Sciences (SPSS) v.28.

## 3. Results

GXT termination time is presented in Figure 1. Comparison results suggest that the time to test termination for the no intervention condition (26.06 min) was significantly higher than for both the N95 (24.20 min, *p* = 0.04) and cloth mask (24.06 min, *p* = 0.03) conditions. However, test termination times for the N95 and cloth mask conditions were not significantly different (*p* < 0.05).

Figure 2 presents the estimated linear physiological and perceived exertion responses over time for each intervention condition. The accompanying Table 2 presents the results of the simple linear regression models for each variable of interest by intervention conditions as well as the interaction model results to compare the regression slopes across the three intervention conditions.

Heart rate increased linearly and significantly over time in all three intervention conditions (*p* < 0.001 for all). No significant differences in the slope of the heart rate response were indicated (*p* > 0.05 for all comparisons). Similarly, respiration rate increased linearly over time for all three intervention conditions (*p* < 0.001 for all). However, comparison of the regression slopes suggests that the positive slope of the no-intervention condition differed significantly from the slope of the cloth mask, where the no-intervention condition had a steeper positive slope than the cloth mask (*p* = 0.04). SpO_2_ increased linearly across time for all three intervention conditions (*p* < 0.001 for all), and no differences in these slopes were indicated between the intervention conditions. TcPCO_2_ decreased linearly over time in the no-intervention condition (B = −0.89, *p* < 0.001); however, the N95 and cloth mask conditions did not significantly change over time. When comparing the slopes between conditions, no intervention differed significantly from the N95 and cloth mask (*p* = 0.02 and *p* = 0.03, respectively). Lastly, RPE increased significantly over time for all three intervention conditions (*p* < 0.001 for all). However, the N95 condition showed a significantly steeper increase in RPE over time compared to no intervention (*p* < 0.001). Similarly, the cloth mask condition slope over time was significantly steeper than no intervention (*p* = 0.05). No difference in RPE slope was found between the cloth mask and N95 conditions.

Table 3 compares the responses to the R-COMFI questionnaire across the N95 and cloth mask conditions. While most R-COMFI responses were not significantly different across intervention conditions, a few significant differences did stand out. Participants reported significantly more facial irritation, itching, and pinching in the N95 compared to the cloth mask (*p* < 0.05 for all). Additionally, nose bridge pinching was rated as significantly higher in the N95 compared to the cloth mask (*p* < 0.001). Sweat and moisture buildup was rated as significantly higher in the cloth mask compared to the N95 (*p* = 0.003).

## 4. Discussion

The purpose of this study was to compare the physiological and performance effects of N95, cloth masks, and no intervention used as source control during graded exercise to maximal exertion in elite athletes. Results suggest that wearing cloth masks or N95s may impair exercise performance as compared to no intervention, especially at higher intensities. No differences in physiological responses were apparent between using cloth masks and N95s as source control products. While wearing facial comfort was generally reported to be better in cloth masks compared to N95s, moisture buildup was also reported to be higher in cloth masks.

Under both the N95 and cloth mask conditions, athletes ran for significantly less time during the GXT compared to no intervention. It is difficult to single out what physiological processes contributed most to the decreased test times in the N95 and cloth mask conditions, and it is likely a combination of several factors, including but not limited to increased inspiratory resistance, hypoventilation, increased CO_2_ retention, and/or higher RPE.

Increased inspiratory resistance with the cloth and N95 conditions could have negatively influenced test performance and decreased test time. Several past studies have demonstrated that treadmill performance times decreased linearly with increased inspiratory resistance [3,11,21,22]. These performance decrements are likely linked to inspiratory resistance via resistance-induced hypoventilation, decreased oxygen consumption, and decreased minute ventilation [11]. In accordance with these expected responses, the rate of increase in respiration during the current study was significantly lower in the cloth mask condition compared to no intervention. While the current study did not measure inspiratory resistance and therefore cannot determine differences in inspiratory resistance between the N95 and cloth mask conditions, resistance can be reasonably assumed to be greater in the N95 and cloth mask conditions compared to the no intervention condition. As such, the shorter exercise times in the N95 and cloth mask conditions compared to the no intervention condition may have been related to higher inspiratory resistance and associated changes in ventilatory mechanics in those two conditions. Importantly, this finding aligns with other previous research reporting decreased exercise time, decreased performance, and increased shortness of breath during graded treadmill running with a cloth mask versus no mask [9]. Additionally, participants indicated significantly greater R-COMFI scores in sweat/moisture buildup during the cloth mask condition compared to N95. This higher level of sweat and moisture buildup in the cloth mask condition likely additionally increased the inhalation resistance resulting in longer, slower respiration rate to maintain the same tidal volume (not directly measured in the current study).

Athletes wearing the N95s and cloth masks also experienced significantly greater rates of increase in RPE over test time compared to no mask. Previous research has pointed to an inverse relationship between exercise duration and rate of increase in RPE [23]. As such, it is possible this faster increase in RPE during both N95 and cloth mask conditions resulted in earlier voluntary exercise test termination. The increased RPE itself may be related to an athlete’s previous perception of exercising while wearing a N95 or mask or potentially from afferent inputs to the brain signaling increased breathing resistance and CO_2_ retention. While it can be presumed that most participants had some previous experience with using and exercising in masks, this study did not systematically collect previous respirator or mask use information from the participants. The impact of previous mask use and the underlying mechanisms should be explored further in the future [24].

Although values did not reach clinical significance, TcPCO_2_ levels in athletes during the N95 and cloth mask condition remained relatively unchanged while exercise intensity increased. This contrasts with the no-mask condition in which TcPCO_2_ significantly decreased as exercise intensity increased. Increased CO_2_ retention is common in other studies examining the use of respirators during exercise [7] and is likely a result of increased exhalation resistance (making it more difficult to exhale CO_2_ past the mask or respirator surface) combined with dead space (the space between the mouth and the inside of the mask or respirator surface) while wearing respiratory protective devices [25]. Specifically, greater dead space of a mask or respirator allows for CO_2_ to be trapped within the mask or respirator and therefore re-breathed by the user, especially during heavy breathing [25]. While not statistically significant, it does appear that the CO_2_ retention in the N95 condition (cup-shaped with high dead space) may be greater compared to the cloth mask (flat mask against the face with lower dead space). This notion has been supported in previous studies with related aims to the current study [8,10]. However, this hypothesis should be explored in further research.

While several previous studies including a systematic review have acknowledged the potential for exercise performance decrements while wearing respirators or other face coverings, the body of literature has also proposed that negative performance impacts are often small and not meaningful to the average wearer [6,7,8,22]. The literature supporting that proposition thus far has been generally focused on moderate- or light-intensity physical activity in occupational health applications. However, when considering N95 or cloth mask use by elite athletes in high-performance situations, the definition of meaningful impact may need to be revisited. This study provides evidence to suggest high-intensity or maximal exercise performance may be impaired by mask wearing in elite athletes. These conclusions support the conclusions of a previous study by Driver et al. who demonstrated a 14% reduction in exercise time on a GXT while wearing a cloth face mask [9]. While most respirator/mask users are not likely concerned with this type of performance, these impacts may in fact be quite meaningful for elite athletes during training and competition and should be considered further.

Considerations for N95 or cloth mask use during exercise should not only include physiological implications but also wearer comfort. While many of the aspects of comfort and tolerability assessed were not reported as being different between the N95 and cloth mask conditions, facial irritation, itching, pinching, and nose-bridge pinching were all rated as being more uncomfortable in the N95 versus the cloth mask. This is unsurprising given that an N95 is designed to interface and seal directly with the wears skin while a cloth mask, by design, does not need to interface with the skin as directly. Inversely, and as mentioned previously, sweat and moisture buildup was reported to be higher in the cloth mask. It was expected that the N95 would have a greater degree of moisture buildup due to the direct interface with the skin trapping moisture, however, it could be that the flat design of the cloth mask brought the moisture on the mask closer to the users face during use, increasing the perception of moisture build up. Given that the physiological responses across the two masks conditions were relatively similar, these differences in comfort may be an important factor for end users to consider when choosing a product to use for source control.

### Strength and Limitations

The authors recognize that the selection and performance of the cloth mask in the current study does not represent the wide and varied types of cloth masks on the market. The cloth masks selected for this study were based on CDC guidelines for selecting a mask (which included having two or more layers of washable breathable fabric) as well as the popularity of cloth mask types used by athletes [26]. Another limitation of the current study is the lack of blinding. Given the nature of the study, neither the researchers nor the participants were blinded to the conditions. An athlete’s bias to wearing an N95 or mask may influence variables such as voluntary exercise test termination time and RPE. However, it should have little impact on physiological variables such as TcPCO_2_, SpO_2_, and ventilatory mechanics. Additionally, in an applied setting, athletes would not be blinded to having to wear an N95 or mask, so any influential perceptions on performance and exertion would still likely be present. The GXT to fatigue used in the current study may not be representative of the physiological demands of different sports (i.e., intermittent high-intensity activity). However, the GXT was selected to measure athlete responses at increasing intensities as well as to help control for variation between conditions. This study did not evaluate the fit or seal of the N95 respirator to the participants’ faces. This check was omitted in preference for external validity where athletes may use N95 respirators for source control without formal fit-testing. However, physiological burden would potentially be increased in the N95 condition if the seal was maintained throughout the exercise. Lastly, the study was limited by its small sample size which could have limited its ability to detect clinically meaningful differences in some of the measured variables, especially those with high variability (e.g., TcPCO_2_, SpO_2_, etc.).

## 5. Conclusions

Wearing cloth masks or N95s may impact exercise performance, especially at higher intensities in elite-level athletes. While no physiological differences were found between cloth masks and N95s, the no-mask condition resulted in increased GXT duration, slower rate of increase in RPE, and steeper rates of decline in TcPCO_2_ as compared to N95 and cloth mask conditions. Although athletic performance was altered while wearing a respirator or mask, it is important to note that the differences in physiological variables did not reach levels of clinical significance for cardiopulmonary dysfunction. As has been concluded in previous studies [22], there is no evidence to suggest that wearing either an N95 or cloth mask during exercise for source control induces any risk to the athlete’s cardiopulmonary health and safety. Thus, the decision of whether an athlete should wear a respirator or mask during exercise should be considered in relation to the potential negative performance outcomes and the mitigation of viral spread and infection.

## Figures and Tables

**Figure 1 ijerph-19-07586-f001:**
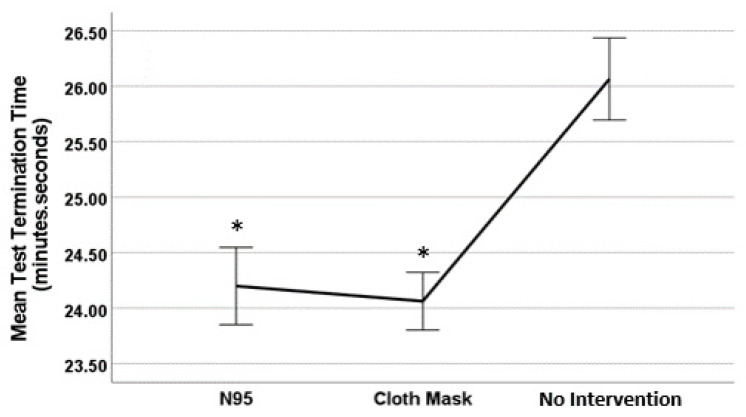
Graded exercise test termination time across three conditions. Note: error bars are 95% CI around the mean. * = significate difference in test termination time compared to no intervention (*p* < 0.05).

**Figure 2 ijerph-19-07586-f002:**
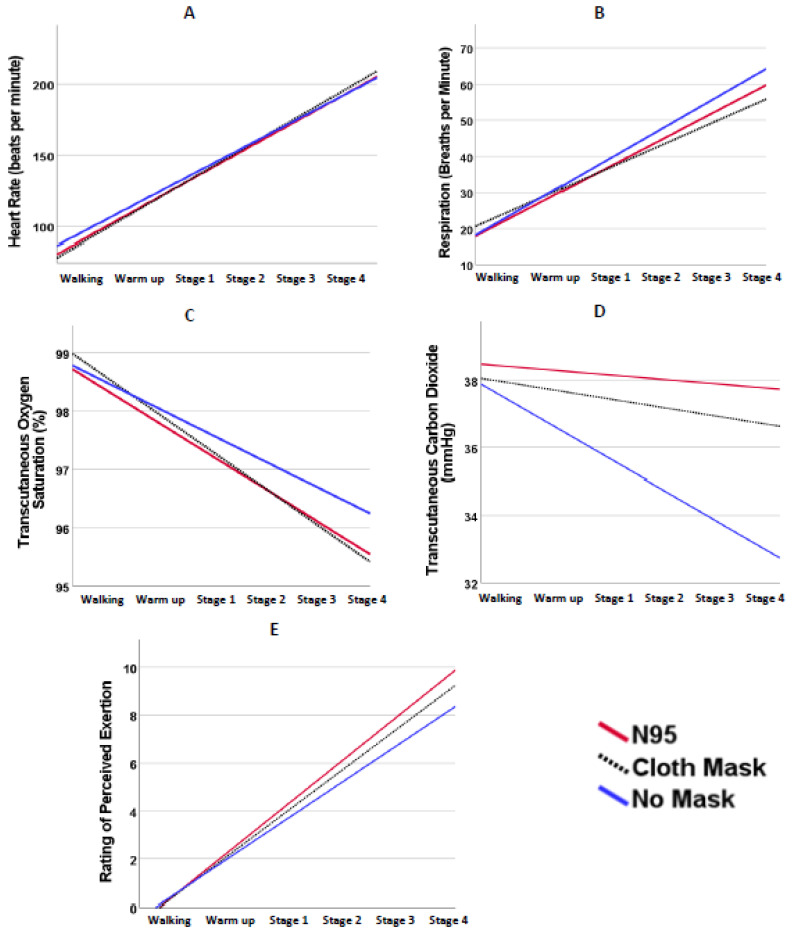
Linear physiological and perceived exertion responses across conditions. (**A**) Heart rate (**B**) Respiration rate (**C**) Transcutaneous oxygen saturation (**D**) Transcutaneous carbon dioxide (**E**) Rating of perceived exertion. Statistical comparisons of the regression slopes can be found in Table 2.

**Table 1 ijerph-19-07586-t001:** Sample characteristics (*n* = 12).

Age (years)	20.1 ± 1.2
Sex	
Female	6 (50%)
Male	6 (50%)
Weight (kilograms)	68.7 ± 9.1
Height (meters)	1.7 ± 0.1
Body Mass Index (kg/m^2^)	23.5 ± 1.6
Sport	
Gymnastics (F) *	2 (16.7%)
Wrestling (M) *	5 (41.7%)
Soccer (F) *	4 (33.3%)
Swimming (M) *	1 (8.3%)

All data are presented as either mean ± standard deviation or frequency (proportion) as necessary. * M and F indicate male or female sports teams/participants.

**Table 2 ijerph-19-07586-t002:** Comparison of linear physiological and perceived exertion responses across conditions.

Dependent Variable	Simple Slope Over Time					*p* Value of Difference in Regression Slope		
	B	Beta	Standard Error	*p* Value	R^2^ (Adjusted)	No Mask vs. N95	No Mask vs. Cloth Mask	N95 vs. Cloth Mask
Heart Rate								
N95	22.67	0.88	1.24	<0.001	0.81	0.54	0.25	0.55
Cloth Mask	23.35	0.88	1.15	<0.001	0.84			
No Intervention	21.10	0.90	1.16	<0.001	0.83			
Respiration Rate								
N95	7.61	0.84	0.57	<0.001	0.76	0.38	0.04	0.19
Cloth Mask	6.20	0.76	0.61	<0.001	0.63			
No Intervention	8.32	0.85	0.58	<0.001	0.75			
SpO_2_								
N95	−0.58	−0.62	0.09	<0.001	0.35	0.36	0.16	0.59
Cloth Mask	−0.64	−0.69	0.08	<0.001	0.52			
No Intervention	−0.47	−0.48	0.09	<0.001	0.32			
TcPCO_2_								
N95	−0.14	−0.07	0.26	0.60	0.14	0.02	0.03	0.72
Cloth Mask	−0.26	−0.13	0.21	0.23	0.19			
No Intervention	−0.89	−0.51	0.18	<0.001	0.33			
RPE								
N95	1.91	0.96	0.05	<0.001	0.92	<0.001	0.05	0.23
Cloth Mask	1.79	0.95	0.06	<0.001	0.90			
No Intervention	1.59	0.92	0.08	<0.001	0.84			

Abbreviations: B = regression coefficient, Beta = standardized regression coefficient, SpO_2_ = transcutaneous oxygen saturation, TcPCO_2_ = transcutaneous carbon dioxide, RPE = rating of perceived exertion. All models controlled for the effect of subject age, height, weight, and BMI.

**Table 3 ijerph-19-07586-t003:** Subjective comfort and wearing experience of the N95 and cloth masks.

Symptom	N95	Cloth Mask	*p* Value
tightness of straps	1.58 ± 0.67	1.33 ± 0.49	-
facial irritation (leaves marks/indents)	1.92 ± 0.67	1.25 ± 0.45	0.009
facial itching	2.00 ± 0.85	1.33 ± 0.49	0.028
facial pinching	1.92 ± 0.79	1.17 ± 0.39	0.008
nose, nose-bridge pinching	2.25 ± 0.75	1.17 ± 0.39	<0.001
facial heat/warmth	2.75 ± 0.45	2.58 ± 0.51	-
sweat/moisture buildup	2.17 ± 0.58	2.83 ± 0.39	0.003
lack of fresh air	2.67 ± 0.49	2.67 ± 0.49	-
nausea	1.17 ± 0.39	1.08 ± 0.29	-
headache	1.17 ± 0.39	1.08 ± 0.29	-
dizziness	1.25 ± 0.45	1.08 ± 0.29	-
loss of energy/tiredness/fatigue	2.08 ± 0.67	2.17 ± 0.58	-
claustrophobia	1.50 ± 0.80	1.58 ± 0.67	-
shortness of breath	2.33 ± 0.49	2.42 ± 0.67	-
difficulty breathing	2.58 ± 0.67	2.58 ± 0.51	-
dry or itchy eyes	1.17 ± 0.39	1.08 ± 0.29	-

All data presented as mean ± standard deviation. All scores are on a scale from 1 to 3 where higher values represent greater experience of the symptom.

## Data Availability

All data will be made available by the corresponding author upon reasonable request.

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
