# Peer review of "Effect of Cloth Masks and N95 Respirators on Maximal Exercise Performance in Collegiate Athletes"

_ijerph, 2022, doi:10.3390/ijerph19137586_

Round 1

Reviewer 1 Report

The manuscript by Darnell et all is well written, appears to be well designed and executed, and is certainly in an area of interest. I appreciate the opportunity to review this paper and make the following suggestions/comments to the authors to improve the work and increase impact for the readership of IJERPH

Two of the authors do not have affiliations (Carpenter, Meglino)

The last sentence of the abstract notes the lack of physiological difference between N95s and cloth masks for outcomes investigated. As a summary statement, suggest including that there were significant physiological differences compared to control.

Has the hexoskin smart shirt been validated? A quick look at their website indicates that there are several works addressing this. Citation(s) and/or mention regarding the validity of this tool for assessment of HR and RR during a GXT would be warranted in the methods.

Methods and statistical analysis are sound. A couple questions:

1) Was the same investigator providing the verbal encouragement for GXTs and was this standardized in any way? 2) With regards to the three GXTs, the randomization and counterbalancing is a positive, but how much time was given between condition trials for the participants? Was this standardized?

Figure 1 could be improved: 1) The image quality appears poor (increase DPI). 2) There is no symbol indicating the significant difference between no intervention and N95 and Cloth Mask

Figure 2 must be improved: 1) Image quality is poor in terms of both resolution and clarity. It is difficult to discern between the conditions. Suggest using color and/or line style (e.g., solid, dashed, dotted) to help differentiate visually. 2) indicate in each panel where significant differences exist as possible and update legend accordingly with reference to Table 2

Suggest addressing the small sample size in the limitations paragraph. While the study was sufficiently powered to detect GXT duration differences between conditions, was there enough power to detect clinically/practical differences in other measures (e.g., TcPCO2).

Reviewer 2 Report

The main aim of the paper „Effect of Cloth Masks and N95 Respirators on Maximal Exercise Performance in Collegiate Athletes“ is to compare the physiological and performance effects of three source control interventions (N95, cloth mask, and no intervention) during high intensity graded exercise in elite athletes.

The study is interesting. I would like to appretiate the efforts of the authors. However, some facts need to be explained.

Major comment:

Page 2, Line 16: This is not fully true. Several studies addressing the effect of respirator at high intensities have been published (I recommend adding this information to the text, also to the discussion), e. g.:

Epstein, D., Korytny, A., Isenberg, Y., Marcusohn, E., Zukermann, R., Bishop, B., Minha, S., Raz, A., & Miller, A. (2021). Return to training in the COVID-19 era: The physiological effects of face masks during exercise. Scand J Med Sci Sports, 31(1), 70-75. https://doi.org/10.1111/sms.13832

Driver, S., Reynolds, M., Brown, K., Vingren, J. L., Hill, D. W., Bennett, M., Gilliland, T., McShan, E., Callender, L., Reynolds, E., Borunda, N., Mosolf, J., Cates, C., & Jones, A. (2022). Effects of wearing a cloth face mask on performance, physiological and perceptual responses during a graded treadmill running exercise test. Br J Sports Med, 56(2), 107-113. https://doi.org/10.1136/bjsports-2020-103758

Fikenzer, S., Uhe, T., Lavall, D., Rudolph, U., Falz, R., Busse, M., Hepp, P., & Laufs, U. (2020). Effects of surgical and FFP2/N95 face masks on cardiopulmonary exercise capacity. Clin Res Cardiol, 109(12), 1522-1530. https://doi.org/10.1007/s00392-020-01704-y

Minor comments:

Figure 2: The image quality is low. In some pictures you can not tell which line represent what. Try to provide pictures with better quality.

Table 2: Please provide more info about the parameter: Respiration. And what does the column marked with the letter B mean?

The first and second paragraphs of the discussion represent results that have already been published in the above mentioned studies.

At the end of the third paragraph of the discussion, tidal volume is mentioned. It has never been mentioned before. Has this parameter been measured and evaluated?

The first 3 lines on page 9 again confirm the already published conclusions.

Round 2

Reviewer 2 Report

The authors have incorporated the requested comments, I have no further requirements.